

# Development of an *In Situ* Dual-Channel Thermal Desorption Gas Chromatography Instrument for Consistent Quantification of Volatile, Intermediate Volatility and Semivolatile Organic Compounds

Rebecca A. Wernis[1,2], Nathan M. Kreisberg[3], Robert J. Weber[2], Yutong Liang[2], John Jayne[4], Susanne Hering[3], Allen H. Goldstein[1,2]

[1]Department of Civil and Environmental Engineering, University of California Berkeley, Berkeley, CA, 94720, USA
[2]Department of Environmental Science, Policy and Management, University of California Berkeley, Berkeley, CA, 94720, USA
[3]Aerosol Dynamics, Inc., Berkeley, CA, 94710, USA
[4]Aerodyne Research, Inc., Billerica, MA, 01821, USA

*Correspondence to*: Rebecca A. Wernis (rwernis@berkeley.edu)

**Abstract.** Aerosols are a source of great uncertainty in radiative forcing predictions and have poorly understood health impacts.
Most aerosol mass is formed in the atmosphere from reactive gas phase organic precursors, forming secondary organic aerosol
(SOA). Semivolatile organic compounds (SVOCs) (effective saturation concentration, $C^*$, of $10^{-1}$–$10^3$ µg m$^{-3}$) comprise a large
fraction of organic aerosol, while intermediate volatility organic compounds (IVOCs) ($C^*$ of $10^3$–$10^6$ µg m$^{-3}$) and volatile
organic compounds (VOCs) ($C^* \geq 10^6$ µg m$^{-3}$) are gas phase precursors to SOA and ozone.

The Comprehensive Thermal Desorption Aerosol Gas Chromatograph (cTAG) is the first single instrument
simultaneously quantitative for a broad range of compound-specific VOCs, IVOCs and SVOCs. cTAG is a two-channel
instrument which measures concentrations of $C_5$–$C_{16}$ alkane equivalent volatility VOCs and IVOCs on one channel and $C_{14}$–
$C_{32}$ SVOCs on the other coupled to a single High Resolution Time of Flight Mass Spectrometer, achieving consistent
quantification across 15 orders of magnitude of vapor pressure. cTAG obtains concentrations hourly and gas–particle
partitioning for SVOCs bihourly, enabling observation of the evolution of these species through oxidation and partitioning into
the particle phase. Online derivatization for the SVOC channel enables detection of more polar and oxidized species.

In this work we present design details and data evaluating key parameters of instrument performance such as I/VOC
collector design optimization, linearity and reproducibility of calibration curves obtained using a custom liquid evaporation
system for I/VOCs and the effect of an ozone removal filter on instrument performance. Example timelines of precursors with
secondary products are shown and analysis of a subset of compounds detectable by cTAG demonstrates some of the analytical
possibilities with this instrument.



## 1 Introduction

In recent years, understanding of organic aerosol (OA) sources has changed substantially. Globally, the burden of secondary organic constituents, i.e. those formed via atmospheric transformation processes, is much larger than that of directly emitted primary organic particulate matter. This is observed in both rural and urban areas (Docherty et al., 2008; Jimenez et al., 2009; Williams et al., 2010; Zhang et al., 2007). Volatile organic compounds (VOCs) are critical precursors to OA. VOC oxidation controls the cycling of hydroxyl and nitrogen oxide radicals and the formation of tropospheric ozone (Atkinson and Arey, 2003). The oxidation of VOCs produces lower vapor pressure compounds that form secondary organic aerosol (SOA) through condensation onto preexisting particles, through new particle formation and growth (Seinfeld and Pankow, 2003) and through aqueous oxidation processes in aerosol or cloud water (Ervens et al., 2011). The fate of approximately half the VOC emissions entering the atmosphere cannot be observationally accounted for, and this discrepancy is likely at least partially due to a lack of comprehensive measurements of speciated organic constituents in a variety of atmospherically distinct environments (Goldstein and Galbally, 2007; Hallquist et al., 2009; Heald and Kroll, 2020).

Recent work has demonstrated that a large fraction of organic aerosol (OA) is semivolatile (Robinson et al., 2007). Intermediate volatility organic compounds (IVOCs, defined as having an effective saturation concentration $C^*$ of $10^3$ to $10^6$ $\mu g\ m^{-3}$) and semivolatile organic compounds (SVOCs, $C^*$ of $10^{-1}$ to $10^3$ $\mu g\ m^{-3}$) have been proposed as a substantial unaccounted for source of SOA in urban areas (Robinson et al., 2007; Weitkamp et al., 2007) but are notoriously hard to measure (Goldstein and Galbally, 2007; Hunter et al., 2017; Isaacman-VanWertz et al., 2018). This hypothesis is supported by estimates of intermediate volatility and semivolatile organic compound (I/SVOC) abundances in the atmosphere that are an order of magnitude larger than primary organic aerosol (Robinson et al., 2007), and direct evidence that oxidation of compounds in the IVOC range efficiently produces SOA (Chan et al., 2009; de Gouw et al., 2011; Lim and Ziemann, 2009; Presto et al., 2010).

There is a need for simultaneous measurement of VOCs, IVOCs and gas- and particle-phase SVOCs with sufficient temporal resolution to track the rapidly changing chemical composition and atmospheric conditions that directly affect SOA formation reactions. Owing to the enormous range of volatility encompassed by VOCs and I/SVOCs, this has typically been achieved through collocation of at least two separate instruments – one to measure VOCs ($C^* \geq 10^6\ \mu g\ m^{-3}$) and IVOCs and another to measure SVOCs. VOCs have traditionally been measured using one of two methods. One way is to collect onto a bed of adsorbent materials and desorb and analyze into a gas chromatograph (GC) coupled to a flame ionization detector (FID) or a quadrupole mass spectrometer (MS) (e.g. Gentner et al., 2012; Goldan et al., 2004; Goldstein et al., 1995; Hopkins et al., 2003; Lamanna and Goldstein, 1999; Lerner et al., 2017; Millet et al., 2005). This method offers excellent chemical specificity as isomers are detected separately and low detection limits as samples are usually collected for many minutes before analysis, but due to the GC temperature ramp and sample collection time has a temporal resolution of 20 minutes to 1 hour. Another widespread method is chemical ionization mass spectrometry (CIMS), e.g. the Proton Transfer Reaction Mass Spectrometer





(Ionicon Analytik) and related technologies, which compared to GC-based methods offers far greater temporal resolution but less specificity as isomers cannot be separated.

Particle-phase SVOCs and lower volatility organics have traditionally been collected on disposable filters and analyzed offline via GCMS (Turpin et al., 2000). Field measurement techniques for gas- and particle-phase SVOCs have been

developed relatively more recently. The FIGAERO-CIMS utilizes automated quartz filter collection for aerosol particles and controlled thermal desorption into a CIMS instrument (Lopez-Hilfiker et al., 2014), offering similar advantages and tradeoffs of CIMS applied to gas-phase measurements. The Volatility And Polarity Separator (VAPS) sacrifices detailed speciation in order to functionally characterize a larger fraction of the organic aerosol mass (Martinez et al., 2016). The Thermal desorption Aerosol Gas chromatograph (TAG) family of instruments, consisting of a reusable filter-based collection cell or impactor cell

coupled to a GCMS, maximizes chemical speciation of gas and particle SVOCs, including separation of isomers, at hourly time resolution. The first TAG was developed by Williams et al. (2006) with the impactor cell, sensitive to particle-phase SVOCs only. Later versions incorporated an automatic liquid injection system for calibrations (Isaacman et al., 2011), a filter cell and denuder for measurement of gas-phase SVOCs and gas–particle partitioning of SVOCs (the SV-TAG; Zhao et al., 2013), a valveless injector for transferring the sample onto the GC column with minimal losses (Kreisberg et al., 2014) and

online derivatization to enable detection of polar SVOCs in addition to nonpolar ones (Isaacman et al., 2014). A version of TAG using the impactor cell, valveless injector and online derivatization is commercially available from Aerodyne Research, Inc. Recently, automated quartz filter collection with thermal desorption and GCMS for chemical speciation of particle-phase SVOCs has also been developed (Cropper et al., 2017; Ren et al., 2019).

In recent years both I/VOC instruments and SVOC instruments have benefitted from field-deployable High

Resolution Time of Flight Mass Spectrometry (HRToFMS, e.g. TOFWERK and IONICON Analytik). This technology affords two main advantages over quadrupole mass spectrometers. First, the high resolution ($m/\Delta m \approx 4000$ or more) allows for identification of chemical formulas of ions detected due to the unique mass defects of the different elements. Second, whereas the quadruple MS scans over a user-specified range of atomic mass units, presenting an implicit tradeoff between sensitivity and the range of possible ions detected (because a smaller range or set of discreet masses to detect allows for increased dwell

times), the HRToFMS has no such limitation and allows the user to detect the entire useful range of ion masses with no compromise in sensitivity and indeed overall enhanced sensitivity.

Until now no single instrument existed that can measure both precursors and their SOA products at the speciated molecular level that spans the relevant 15 decades in volatility. The Comprehensive Thermal desorption Aerosol Gas chromatograph (cTAG) combines a I/VOC collector based upon the design of Gentner et al. (2012) and one channel of the

SV-TAG joined together before a HRToFMS to access this entire range of organic volatility at once.


## 2 Description of instrument

Utilizing reusable adsorbent and stainless steel filter collection, thermal desorption, online derivatization, GC and HRToFMS in a dual-channel setup the cTAG measures concentrations of speciated organic compounds from $C_5$ through $C_{32}$ alkane equivalent volatility. The instrument operates in an automated fashion with hourly time resolution for concentrations and gas-

particle partitioning measurements of SVOCs bihourly. Derivatization of SVOCs allows for detection and analysis of polar compounds with hydroxyl groups in addition to nonpolar species. In this section, we describe the key components of the system, the sample collection, separation and detection pathways, calibration methods, timing, and data processing procedures. Figure 1 shows a schematic of the instrument.

### 2.1 Inlet and I/VOC channel

Ambient air enters the instrument through either a BGI sharp cut $PM_{2.5}$ cyclone or a BGI sharp cut $PM_1$ cyclone depending on application-specific criteria (SCC BGI Inc., Waltham, MA) (Kenny et al., 2000). The flow is then split to collect I/VOCs and SVOCs in parallel.

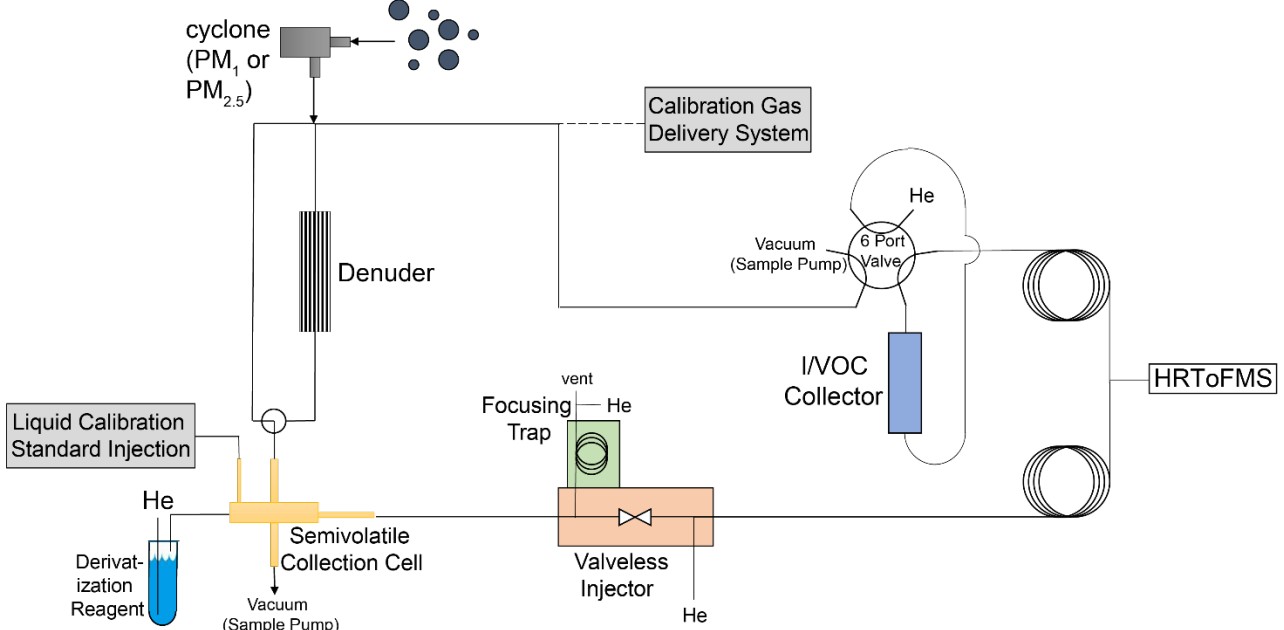

**Figure 1** Schematic of cTAG. VOCs and IVOCs are collected on the I/VOC collector while SVOCs are collected on the collection and
thermal desorption cell. The I/VOC and SVOC channels are independent until the GC column outputs from each channel meet and enter the HRToFMS. Optionally, the SVOC channel can collect only particle-phase SVOCs by first passing sampled air through a denuder that removes all gas-phase compounds. A derivatization agent introduced upon desorption of the SVOCs from the collection cell enables detection of polar SVOCs in addition to nonpolar ones. Sample collection happens in parallel, for 23 minutes, followed by analysis of the I/VOC collector contents, then analysis of the semivolatile collection cell contents, for a total turnaround time of one hour. Calibration is
done for VOCs and IVOCs by manually disconnecting the ambient inlet and connecting and sampling the output of the calibration gas delivery system (Fig. 3) and for SVOCs via liquid standard injections onto the semivolatile collection cell.



On the I/VOC channel, 50 sccm of ambient air is pulled from downstream of the cyclone through a stack of three quarter-inch diameter punches of sodium thiosulfate ($Na_2S_2O_3$) impregnated glass fiber filters to remove ozone, followed by a six-port valve (6PV, Valco Instruments Co. Inc.) and the I/VOC collector for I/VOC pre-concentration. The I/VOC collector consists of a layered bed of adsorbents based on the design of Gentner et al. (2012) whose types and quantities were chosen in order to efficiently collect I/VOCs with volatilities between those of n-pentane and n-hexadecane in a 1 L total sample volume when the collector is held at 30 °C. Adsorbents are layered from least to most adsorptive strength, with the IVOCs adsorbing on the least adsorptive material and the most volatile VOCs passing through and adsorbing onto the most adsorptive material. Figure 2 is a diagram of the collector. In order along the sample flow path are 60 mg glass beads (Alltech, 60/80 mesh, DCMS-treated), 10 mg Tenax TA (Supelco, 60/80 mesh), 10 mg glass beads, 20 mg Carbopack B (Supelco, 60/80 mesh), 10 mg glass beads, 20 mg Carbopack X (Supelco, 60/80 mesh) and 10 mg glass beads. The glass beads do not efficiently trap VOCs or IVOCs and serve solely as separators for the adsorbents. Once assembled and before installation on the instrument, the collector is conditioned at 325 °C with 50 sccm nitrogen for 3 hours. Breakthrough volume measurements described in Sect. 3.1 were performed to ensure that this bed composition would fully collect and transfer all VOCs and IVOCs in the range of interest.

Adsorbents are packed in a custom, pure stainless-steel housing consisting of a 3.18 mm outer diameter (OD) thin-walled (0.13 mm) tube brazed to a 1.59 mm OD, 0.51 mm ID tube (Fig. 2). A section of the 3.18 mm portion that contains the adsorbents is flattened to 1.59 mm OD to reduce swept volume in the collector and improve heat transfer to enable sharper injections of highly volatile compounds. Brazing to the 1.59 mm tube eliminates the need for a union and the associated internal volume, again aiding the rapid injection of the most volatile species. A stainless steel microfiber mesh screen (Bekaert) installed at the braze point keeps out particles and retains the adsorbents on the upstream sampling end, while a glass wool plug retains the adsorbents on the downstream end. The entire housing is chemically passivated (Inertium® treatment, AMCX, PA, USA) to prevent active compounds from reacting with the stainless steel.

During sampling the I/VOC collector is held at a temperature at least several degrees above the dew point to avoid water condensation in the system; this is typically around 30 °C. After sampling the I/VOC collector is briefly purged with helium to reduce the amount of air and water sent to the detector. Then the 6PV actuates and helium flows through the collector in the reverse direction as the collector heats from ambient temperature to 260 °C to desorb the analytes onto the GC column (metal MXT-624, 30 m, 0.32 mm ID, 1.8 μm phase). Two 100 W cartridge heaters mounted in an aluminum block clamped to the collector housing ensure a rapid initial heating time of 35 s, producing sharp chromatography peaks even for the unretained, most volatile species. Total desorption time is 4 minutes. GC analysis time is 25 minutes, with an initial hold for 1 minute at 40 °C followed by a 10 °C min$^{-1}$ ramp to 250 °C and a 3 minute hold at 250 °C.

## 2.2 SVOC channel

After splitting off from the combined ambient air flow through the cyclone, 10 Lpm of ambient air passes through a stainless steel filter collection and thermal desorption cell (F-CTD) maintained at 30 °C. The F-CTD quantitatively collects compounds





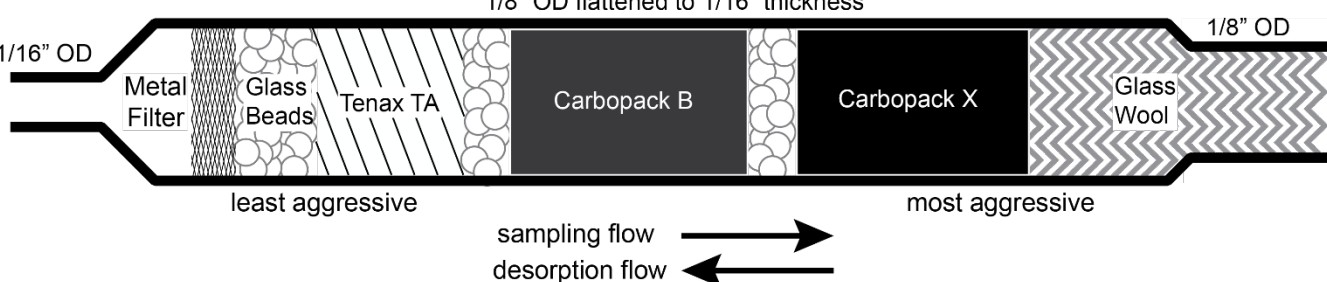

**Figure 2** Schematic of I/VOC collector. Semivolatile and lower volatility species, including particles, are captured on the metal filter. IVOCs are captured on the glass beads and Tenax TA, while the most volatile analytes including isoprene are retained by the more aggressive adsorbents. Flow is reversed for desorption so that the less volatile compounds never interact with the more aggressive adsorbents.

with volatilities as high as n-tetradecane ($C_{14}$) and as low as n-dotriacontane ($C_{32}$) in the gas and particle phases. Directly upstream of the F-CTD, an optional 500-channel activated carbon denuder (MAST Carbon) efficiently removes gas-phase species, allowing for gas-particle partitioning measurements by difference with and without the denuder in line. F-CTD and denuder characteristics are described in further detail in Zhao et al. (2013).

After collection, the F-CTD is heated to 315 °C and flushed with 20–150 sccm helium during a two-stage desorption process as described in Zhao et al. (2013). Upstream of the F-CTD the helium may optionally be bubbled through a liquid reservoir of N-methyl-N-(trimethylsilyl) trifluoroacetamide (MSTFA, Sigma-Aldrich, > 98 % purity, synthesis grade) for online derivatization of species containing hydroxyl groups. Evaluation of online derivatization for TAG is discussed in Isaacman et al. (2014). The sample is re-concentrated onto a focusing trap held at 30 °C made of a 1 meter metal, thick phase chromatography column (MXT-5, 0.53 mm ID, 5 μm phase thickness; Restek). This step allows for faster transfer of low volatility species and efficient purging of excess derivatization agent and by-products. Flow is reversed on the trap to about 2 sccm for transfer of the sample onto the GC column (metal MXT-5, 20 m, 0.18 mm ID, 2 μm phase thickness; Restek) via the restrictive section of a valveless interface (VLI), described by Kreisberg et al. (2014). GC analysis time is 19 minutes, with an initial hold for 1 minute at 50 °C followed by a 20 °C min$^{-1}$ ramp to 330 °C and a 4 minute hold at 330 °C.

## 2.3 Miniature gas chromatographs

cTAG requires two separate GC columns each optimized for the separation of target species on a given channel and whose temperatures are controlled independently. In order to achieve this while preserving the instrument's compactness and time resolution we developed miniature gas chromatographs. Each chromatograph consists of a custom machined aluminum hub around which the metal column is wrapped in a single layer so that it is in thermal contact with the hub along its entire length. On the inside surface of the hub, an expanding split-band 150 W heater heats the hub evenly around its circumference. A thermocouple inserted into a bored hole in the hub body tracks its temperature. PID heating control allows for programmable, reproducible temperature ramping for GC analysis. When the temperature program completes, a fan blows ambient air onto



the hub to bring it back to its initial temperature in time for the next sample injection. As with traditional GC systems, the front of the column may be trimmed (at the expense of one full winding on the aluminum hub) to extend column life. Design schematics, photos and temperature ramp reproducibility data can be found in Appendix A.

5       The use of miniature gas chromatographs has obvious benefits for instrument compactness and field portability. Additionally, the independent temperature control allows analysis on one channel to start while the column from the other channel is cooling down, increasing the maximum possible time resolution as compared with the traditional approach of using a single oven for dual column GC methods. While commercial miniature GC systems are available (e.g. Valco Instruments Company Inc., 2020), we chose to design our own for greater flexibility and the ability to use off-the-shelf columns.

## 2.4 High resolution time of flight mass spectrometer

10   The two chromatography columns are joined in a passivated tee connected to an approximately 18 cm passivated stainless steel 794 mm OD, 125 µm ID tube that serves as the mass spectrometer transfer line, held at a constant 275 °C. While one column progresses through its temperature program, the other is held at its initial temperature with constant flow and the analytes and carrier gas are pulled through the transfer line into the vacuum chamber of the mass spectrometer.

      Chromatographically separated analytes are detected by a field portable HRToFMS (TOFWERK). The HRToFMS is 15  operated at 70 eV electron impact ionization over a user-selected mass-to-charge ratio range, typically from 15 to 450 on cTAG. This allows for matching to compounds found in mass spectral databases. For compounds not identifiable by database matching or authentic standard calibration, the mass to charge resolution of $m/\Delta m \approx 4000$ enables determination of the molecular formula of the ions, lending insight to molecular structure and ultimately compound identity, a critical capability for studying the complex interactions of organic molecules in the atmosphere.

## 20  2.5 Calibration

Calibration on both channels relies on injection and analysis of a suite of authentic standards in varying amounts. On the I/VOC channel, a mixture in either liquid or gas form is introduced into a custom dynamic dilution system for fine control of the output concentration. The dynamic dilution system uses a heated platinum catalyst to generate zero air at ambient relative humidity and a series of mass flow controllers (0–20 and 0–1000 sccm MFCs, Bronkhorst EL-FLOW) and valves to choose 25  and dilute the input mixture for sampling (Fig. 3). Dilution ratios ranging from 50:1 to 1000:1 ensure near ambient relative humidity for diluted mixtures. From the point of mixing to the collector, all plumbing is heated to at least 55 °C. A specialty gas cylinder with multiple components at ppm levels is introduced in one port of the system.

      Calibration using custom compressed gas cylinders limits the user to compounds amenable to storage in cylinders over years and does not allow significant flexibility after standard cylinders are produced. We developed a liquid evaporation 30  system modeled after Jardine et al. (2010) to circumvent these limitations by allowing the user to purchase individual pure standards and prepare liquid solutions for evaporation and sampling. The liquid mixture is drawn into a syringe pump (TECAN Cavro Centris Pump, 250 µL glass syringe) and dispensed into another port of the dynamic dilution system at a rate of 1.6 µL



min$^{-1}$ into a dedicated evaporation chamber. From one end of the chamber a dedicated MFC flows 1000 sccm zero air over the emerging liquid feed. Most of the resulting evaporated mixture in zero air is exhausted while 0–20 sccm is subsampled for subsequent dilution in the same manner as a gas cylinder. Once diluted, the gas or liquid calibration mixture is collected on the I/VOC collector, then desorbed and analyzed exactly like ambient air samples.

5    On the SVOC channel, liquid calibration mixtures are held in reservoirs pressurized under helium. A multiport selector (Rheodyne MHP7970-500-4) selects a reservoir to fill a 5 µL sample loop. A 6-port valve (Rheodyne MHP9900-500-1) actuates to allow pressurized helium to push the liquid out of the loop and inject it directly onto the F-CTD via a dedicated port in the cell housing. A detailed description of the SVOC liquid calibration injection system may be found in Isaacman et al. (2011).

10    Prior to applying calibrations from analysis of authentic standards, ambient data are normalized by an internal standard introduced onto the sampling medium of each channel on every ambient sample. For the I/VOC channel, several ambient long-lived and therefore atmospherically well-mixed anthropogenic compounds with no significant current emission sources serve as suitable internal standards to control for run-to-run variability. Carbon tetrachloride is commonly used for this purpose (Lerner et al., 2017) and CFC-113 and 1,1,1-trichloroethane are also adequate for cTAG on the timescale of a

15    typical field campaign (Engel et al., 2018; Karbiwnyk et al., 2003). The compound with the clearest signal and fewest

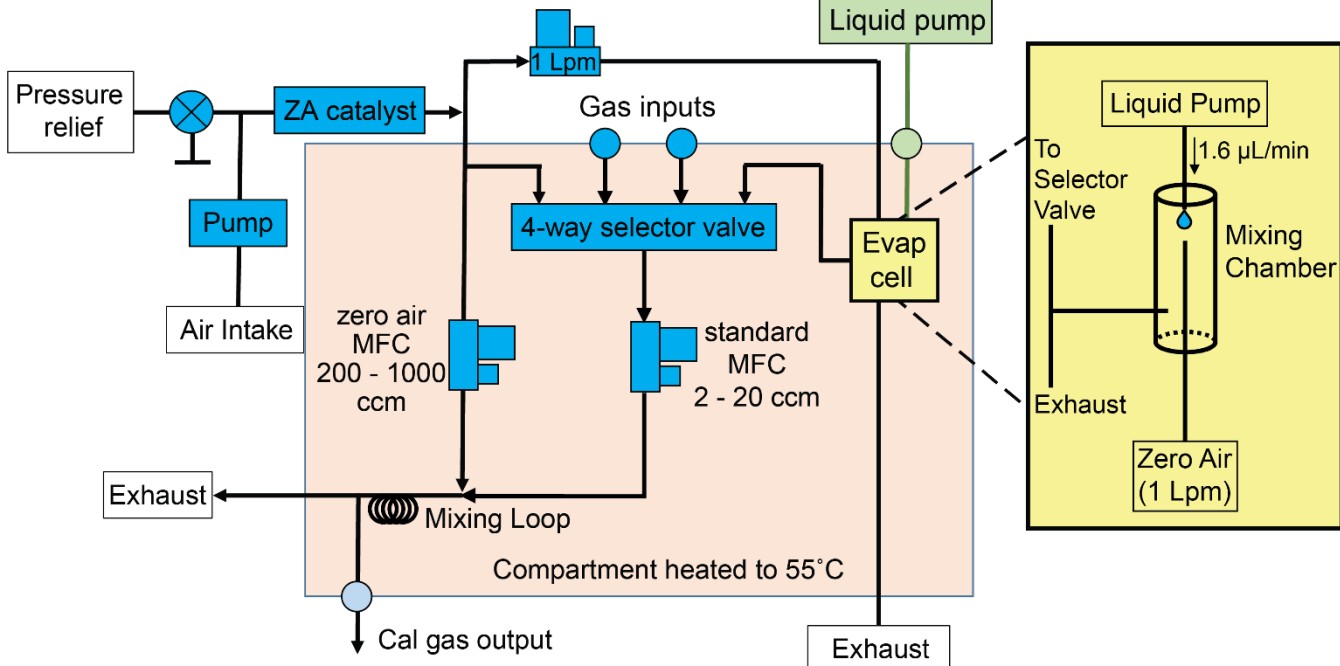

**Figure 3** Schematic of dynamic dilution system for VOC and IVOC calibration. Gas cylinders or liquid mixtures can be used for calibration. Liquid mixtures are introduced into an evaporation cell using a syringe pump. The liquid is evaporated into 1 Lpm of zero air to reach ppm or lower concentrations. The resulting gaseous mixture (or, by user choice, the output of a calibration gas cylinder) is subsampled with a 2–20 ccm MFC and further diluted up to 500 times with more zero air before being sampled by the I/VOC collector. Zero air for the dilution system is generated by passing ambient air through a heated platinum catalyst.



coelutions among these is generally preferred. On the SVOC channel, a single 5 µL loop of a calibration mixture of isotopically labeled compounds spanning a variety of volatilities and functional groups is injected onto the F-CTD after collection of every ambient sample. These compounds are desorbed and analyzed with the sample and serve to track variability of derivatization efficiency and instrument response.

## 2.6 Instrument operation

cTAG is fully automated. All valves, fans, temperature-regulated zones and electronic pneumatic controllers are controlled by a microprocessor-based control box developed by Aerodyne Research, Inc. for commercially available TAG systems and adapted and upgraded for our system. A Microsoft Visual Basic .NET software program operating on a PC platform interfaces between the user and the control box, using serial communication to send commands and load sequences for unattended field operation and receive, display, plot and record temperature and voltage readings.

Total turnaround time for a single instrument cycle is exactly one hour. A typical operation sequence begins with 23 minutes of concurrent sampling on the I/VOC and SVOC channels with the SVOC channel optionally sampling through the denuder to remove gas phase species. During sampling the SVOC chromatographic analysis of the previous hour's sample finishes. Once sampling is completed, the I/VOC collector is purged with helium for 1 minute. Then the 6PV actuates, at which point the 25 minute I/VOC GC temperature program and HRToFMS data acquisition begin and the I/VOC collector is rapidly heated to 280 °C and held at that temperature for 4 minutes while 1.5 sccm of helium carries the analytes to the head of the GC column.

While the I/VOC sample is being analyzed, the SVOC liquid calibration system injects 5 µL of the internal standard mixture onto the F-CTD. The F-CTD is then heated to 315 °C under 20 sccm of helium. Total helium purge flow is controlled by a mass flow controller (MFC). The flow is split downstream of the MFC, with 80 % bubbling through the reservoir of liquid derivatization agent before rejoining the other 20 % and purging the F-CTD. This flow ratio was determined to be sufficient for complete derivatization of all compound classes of interest in Isaacman et al. (2014). After 8 minutes, the total flow is increased to 150 sccm for 4 minutes to aid the transfer of the least volatile compounds. During this process analytes are refocused on the focusing trap, held at 30 °C. Flow is then reversed and the trap heats to 315 °C as the I/VOC GC analysis finishes. Continued purge flow from the MFC raises the pressure in the F-CTD and forces the analytes through the restrictive portion of the VLI and onto the head of the GC column over the course of the 4-minute trap desorption period. The SVOC GC analysis begins, followed by sampling for the next cycle.

A typical field campaign day sees round-the-clock sampling and analysis as previously described, alternating sampling with and without the denuder inline on the SVOC channel. Approximately every 2 weeks, calibrations are performed on both channels of the instrument simultaneously lasting 10 to 12 hours to generate multiple data points per compound for a range of loadings. The stack of sodium thiosulfate filters is replaced every 2 weeks to avoid ozone breakthrough by a safe margin (Sect. 4.4). At least once per field campaign a tank of nitrogen or zero air is plumbed into the inlet and sampled on both channels in order to quantify contaminants in the system.



## 2.7 Data processing

Chromatogram integrations are performed in TERN, software developed in Igor Pro 7 (Wavemetrics) by Isaacman-VanWertz et al. (2017). TERN enables automatic batch integration of single compounds on hundreds of chromatograms at a time for rapid generation of compound concentration timelines. Compounds are organized in templates and the ability to search
commercial or custom mass spectral libraries for compound identification is built in.

## 3 System evaluation

The focus of development and system evaluation was on the I/VOC channel, as the SVOC channel is identical to a single channel of the previously developed SV-TAG instrument extensively documented elsewhere (Isaacman et al., 2011, 2014; Kreisberg et al., 2014; Zhao et al., 2013). Development for the cTAG focused on optimizing the design of the I/VOC collector,
developing a calibration system for the I/VOC channel capable of using custom liquid mixtures, ensuring removal of ozone for accurate collection of ozone-reactive VOCs and IVOCs, and field deployment in a polluted urban area.

## 3.1 I/VOC collector breakthrough tests

The target collection range for the I/VOC channel was chosen to include isoprene, a key biogenic $C_5$ hydrocarbon, on the high volatility end. The low volatility end was chosen to overlap with the SVOC channel range (which starts at $C_{14}$) by several
carbon numbers and to include sesquiterpenes, a class of $C_{15}$ compounds also of biogenic origin but with greatly varying structures and reactivity and which are far less well documented and understood (Bouvier-Brown et al., 2009; Chan et al., 2016; Yee et al., 2018).

        In order to decide the final quantities of adsorbents in the I/VOC collector, three different collector compositions were tested for breakthrough of the most volatile species. Briefly, breakthrough volume is defined as the volume of carrier gas
required to purge an analyte through the adsorbent bed, dependent on adsorbent quantity, temperature and analyte volatility (Definition of Breakthrough Volumes, 2020). The breakthrough volume must be larger than the volume of air sampled for the most volatile analyte of interest to ensure complete collection of all analytes.

        Breakthrough volumes were measured using a real-time VOC instrument, a Proton Transfer Reaction Mass Spectrometer (PTR-MS, IONICON Analytik). The output of a specialty gas cylinder of biogenic VOCs was diluted using the
custom dynamic dilution system to deliver ppb-level concentrations of analytes to the collector, held via PID temperature control at 30 °C. The PTR-MS inlet was connected downstream of the prototype collector while sampling from the controlled gas mixture at 100 sccm. For each analyte, the PTR-MS measured zero concentration until the breakthrough volume was reached, after which the downstream measured concentration rapidly rose and plateaued. The beginning of the rapid rise in concentration was taken to be the point of breakthrough. Based on the results of these tests, a final collector composition was
chosen with quantities of the more aggressive adsorbents in between those of the second and third prototypes tested.



## 3.2 Dynamic dilution system testing

The dynamic dilution system for evaporation of liquid I/VOC mixtures and dilution and delivery of liquid and gas calibration mixtures was evaluated for linearity and reproducibility of delivered concentrations. In addition, for compounds present in both gas cylinders and liquid solutions calibration curves were compared for agreement. Calibrant compounds from liquid (prepared in lab) and gas cylinder (Custom mixture prepared by Apel-Riemer Environmental, Inc., 2019) mixtures were introduced at 0, 1, 2, 4, 8 and 12 ppb (10 times lower concentration for β-caryophyllene) with a total volume of air sampled of 280 cm$^3$ with 6 (0, 1, 2, 4, 8 ppb) or 3 (12 ppb) replicates at each level. From a separate gas cylinder, 1 ppm neohexane was introduced at a constant 1 ccm and used to normalize the responses from the calibrant compounds. This step served to account for variations in sample volume and instrument sensitivity between samples.

## 3.3 Limits of detection

Limits of detection (LOD) are species dependent. On the I/VOC channel, limits of detection were estimated using the following formula:

$$LOD = 3 * \sigma_{blank} / m \tag{1}$$

where $\sigma_{blank}$ is the standard deviation of the integrated area of the chromatographic blank signal (12 replicates) of the quantification ion at the retention time of the analyte in question in units of counts and $m$ is the instrument sensitivity for that analyte, i.e., the slope of the best fit to the calibration curve, in units of counts pptv$^{-1}$ (Foley and Dorsey, 1984).

The SVOC channel is equivalent to one channel of the SV-TAG as described in Isaacman et al. (2014). The average limit of detection reported for that instrument is 1 to 2 ng m$^{-3}$ (~0.1 ppt) for a 20 minute sample collected at 10 Lpm. We determined that the LOD for the SVOC channel on cTAG determined according to equation (1) is consistent with that reported previously for SV-TAG.

## 3.4 Evaluation of ozone removal on the I/VOC channel

Ozone has been shown to be able to penetrate standard VOC sampling inlets and react with certain analytes collected on adsorbent beds like ours (Calogirou et al., 1996; Helmig, 1997; Pollmann et al., 2005). A common practice for ozone removal is to pass the sample flow through a sodium thiosulfate impregnated filter upstream of the adsorbent bed (Helmig, 1997). Pollmann et al. (2005) demonstrated that this technique is effective for some sesquiterpenes as well as the previously studied monoterpenes and hydrocarbons. We explored the effects of ozone on the recovery of a suite of compounds of interest known to have short lifetimes with respect to reaction with ozone, including monoterpenes and sesquiterpenes, and tested how placement of sodium thiosulfate (Na$_2$S$_2$O$_3$) impregnated filters upstream of the collector altered the recovery of these compounds. The Na$_2$S$_2$O$_3$ filters were obtained from the Barsanti research group at UC Riverside (Hatch et al., 2017).

To quantify the effects of the presence of ozone in sampled air, we diluted air from the output of an ozone generator (Model 1008-RS, Dasibi Environmental Corp) with humidified zero air to achieve an ozone concentration of 100 ppb ozone.



We then verified that the ozone was completely removed by three inline filters by measuring the ozone concentration downstream (Model 202, 2B Technologies).

After verifying the efficient removal of ozone by the filter stack, we combined 9 Lpm of 100 ppb ozone with the 1 Lpm output from our dynamic dilution calibration system described in Sect. 2.5. Using that calibration system, we evaporated

a liquid mixture of sesquiterpenes and other compounds to achieve a concentration of 5.60 ppb, which once combined with the 9 Lpm zero air was reduced to 560 ppt for sampling.

We sampled this liquid mixture onto the I/VOC channel under four different conditions: without ozone and without a $Na_2S_2O_3$ filter, without ozone and with the filter, with 100 ppb ozone and without the filter, and with 100 ppb ozone with the filter. This allowed us to establish (1) whether an ozone removal method is needed in our instrument for accurate quantification

of ozone-reactive species such as some sesquiterpenes and (2) whether the presence of the filter altered the concentrations of non-ozone-reactive VOCs and IVOCs or of reactive VOCs and IVOCs in the absence of ozone.

## 3.5 Region of sensitivity overlap

Some of the chemical compounds with alkane equivalent volatility between that of n-tetradecane and n-hexadecane can be detected and quantified on both the I/VOC and the SVOC channels. Comparing quantifications of such compounds between

channels can serve as a useful check of instrument performance. However in practice the number of compounds suitable for such an analysis is small, and for any given ambient data set there may be none. One reason for this is the 200 times larger sample volume collected on the SVOC channel, so that some compounds that are easily quantifiable on the SVOC channel are below detection limit on the I/VOC channel. Compounds which are reactive with ozone on timescales of under an hour or so, including many sesquiterpenes (Fig. 5), must be quantified on the I/VOC channel where ozone is removed at the inlet. Highly

polar species may not elute on the I/VOC column; the SVOC channel with its online derivatization is more appropriate in this case. Thus there is usually reason to choose one channel over the other for quantification purposes.

A survey of compounds from the ambient data set from the cTAG deployment in McCall, Idaho for the 2019 Fire Influence on Regional to Global Environments and Air Quality (FIREX-AQ, 2019) field campaign was conducted to test cross-channel agreement for compounds in the volatility overlap region. Of the compounds found, all but one, bornyl acetate, was

subject to one of the restrictions listed above. Results for bornyl acetate are presented in Section 4.5.

## 3.6 Measurements of ambient air

The cTAG was deployed in Livermore, CA in April 2018 for its first field test. This test was undertaken to evaluate the stability of instrument performance during round-the-clock automated operation and demonstrate the capability to measure concentrations of a variety of compound classes from $C_5$ to $C_{32}$ alkane-equivalent volatility. The instrument was deployed

approximately 100 yards from a Bay Area Air Quality Management District monitoring site to make use of their air quality measurements for comparison (specifically, ozone and ambient temperature reported here). Livermore is a valley city on the eastern and downwind edge of the San Francisco Bay Area, where pollution outflow from the region combined with optimal





conditions for secondary photochemical smog formation frequently leads to the highest ozone levels in the Bay Area (Knoderer et al., 2018). In addition, Livermore has substantial wood burning for heat during winter. An analysis of a subset of VOCs, IVOCs and SVOCs measured in Livermore by cTAG is presented to demonstrate some of the analytical capabilities of this instrument.

| Collector No: | 1 | 2 | 3 | 4 |
|---|---|---|---|---|
| **Bed Material (mg of sorbent)** | | | | |
| Glass beads | 40 | 40 | 40 | 40 |
| Tenax TA | 20 | 20 | 20 | 10 |
| Carbopack B | 30 | 20 | 10 | 20 / 10 |
| Carbopack X | 40 | 30 | 15 | 20 |
| **Breakthrough volume at 30 °C (L)** | | | | |
| MVK+MACR | 13.2 | 8.7 | 2.0 | >2 |
| Isoprene | 18.3 | 19.9 | 3.6 | >2 |

**Table 1** Composition of different I/VOC collectors and corresponding breakthrough volumes for some of the most volatile target compounds. Collector #4 is a compromise between collectors #2 and #3 with reduced Tenax TA.

## 4 Results

### 4.1 I/VOC collector breakthrough tests

Breakthrough volume measurement results for isoprene and the sum of methyl vinyl ketone and methacrolein (MVK+MACR) for several collector prototypes are shown in Table 1. For compactness and to reduce thermal mass to ensure rapid desorption, we targeted a final design containing the minimum quantity of adsorbents that would safely and robustly meet our requirements for breakthrough volume. Collector #3, which had the lowest quantities of adsorbents of the three different prototype collectors tested, had a breakthrough volume of twice our usual sample volume for MVK+MACR. We therefore settled on a fourth prototype with quantities intermediate between collectors #2 and #3 of the most aggressive adsorbents to ensure a safe margin. We also reduced the quantity of Tenax TA, as the 10 mg final quantity provides more than sufficient breakthrough volumes for the lower volatility compounds that it collects (Tenax® TA Breakthrough Volume Data, 2020).

### 4.2 Dynamic dilution system testing

Figure 4 shows calibration curves for benzene, o-xylene, α-pinene and β-caryophyllene from both a standard gas cylinder and a custom liquid solution evaporated and delivered via the dynamic dilution system. Normalized detector response on the y-axis is the integrated area of the chromatographic peak divided by that of neohexane, our internal standard (Sect. 3.2), on the



same chromatogram. Additionally, for benzene all data points had the mean of the 0 ppb values subtracted from them to account for a small persistent benzene contaminant likely due to the presence of Tenax TA in our adsorbent bed (Cao and Hewitt, 1994). Points are mean values of the six individual samples at each level (three samples at the highest level) with error bars representing plus or minus one standard deviation. To evaluate performance against our theoretical model of linear

response with zero response at zero concentration, dotted curves are best fit lines forced through the origin.

These compounds show excellent linearity with $R^2 \geq 0.93$. Agreement between gas and liquid is within 5 %, as measured by the difference in the slopes of the best fit lines. The compounds span a range of volatility from that of a $C_6$ alkane to a $C_{15}$ alkane, demonstrating the dynamic dilution system's suitability for evaporating and diluting IVOCs as well as VOCs.

Isoprene and methyl vinyl ketone were also analyzed and found to be linear, with $R^2 \geq 0.92$. However, we found

consistently and significantly lower responses using the liquid mixtures, likely indicating issues during preparation and storage of the liquid solution related to the high volatility and reactivity of these compounds. For $C_5$ to $C_6$ alkane-equivalent volatility compounds, we thus plan to continue to rely on gas cylinder standards until and unless our liquid preparation process can be modified to remove this source of error.

Octamethylcyclotetrasiloxane and decamethylcyclopentasiloxane also had excellent linearity ($R^2 \geq 0.91$) but were

found to have about 30 % higher responses using the liquid solution than using the gas cylinder. This could suggest depletion of these compounds in the gas cylinder, but we were not able to test this hypothesis.

### 4.3 Limits of detection

Limits of detection were estimated for the 44 compounds present in the Photochemical Assessment Monitoring Stations 57-component commercial standard (Scott-Specialty) that are within the volatility range collected on cTAG's I/VOC channel,

including linear, branched and aromatic hydrocarbons (Table 2). The LODs range from 0.5 to 8 pptv except for benzene and toluene which have known contamination issues from the Tenax used in the collector (Cao and Hewitt, 1994).

### 4.4 Evaluation of ozone removal on the I/VOC channel

At 1 Lpm (20 times our ambient sampling rate), ozone breakthrough through a stack of three $Na_2S_2O_3$ filters occurred after 8 hours, indicating a single filter stack can be used without breakthrough for 9,600 minutes, or about 20 days of continuous

operation.

Figure 5 shows the effects of the $Na_2S_2O_3$ filter on several analytes of interest with and without ozone. In the absence of ozone, VOC and IVOC concentrations were unperturbed by the presence of the filter except for manageable losses of lower volatility compounds when the filter was present (Fig. 5 (a)). In the presence of ozone, placement of the $Na_2S_2O_3$ impregnated filters in the sampling flow upstream of the collector prevented degradation (Fig. 5 (b)). In general, losses of analytes in the

absence of the filter were greater for compounds with lower atmospheric lifetimes with respect to reaction with ozone.

The results imply having a $Na_2S_2O_3$ impregnated filter inline improves quantification of ozone-reactive species without significant downsides. However, prior literature suggests some very polar compounds, not tested in this experiment,



**Figure 4** Calibration curves with $R^2$ values for (a) benzene, (b) o-xylene, (c) α-pinene and (d) β-caryophyllene in a gas cylinder (red) and in a custom liquid mixture (blue), delivered by the dynamic dilution system (Fig. 3). Normalized detector response on the y-axis is the integrated area of the chromatographic peak divided by that of neohexane, our internal standard, on the same chromatogram. Maximum normalized detector response for a given compound ranged from 0.030 (β-caryophyllene, max concentration delivered 1.2 ppb) to 3.0 (o-Xylene, max concentration delivered 12 ppb). Additionally, for benzene all data points had the mean of the 0 ppb values subtracted. Points are mean values of the individual samples at each level with error bars representing plus or minus one standard deviation. Dotted lines are best fit lines forced through the origin, while $R^2$ values are from best fits not forced through the origin. Compounds show excellent linearity and agreement within 5 % between the gas cylinder and liquid solution.


| Compound | LOD (pptv) | Compound | LOD (pptv) | Compound | LOD (pptv) |
|---|---|---|---|---|---|
| Isopentane | 2.0 | 2,3-Methylpentane | 3.8 | o-Xylene | 3.7 |
| 1-Pentene | 0.5 | 3-Methylhexane | 0.6 | Styrene | 7.1 |
| Pentane | 2.2 | Cyclohexane | 0.9 | Cumene | 1.1 |
| trans-2-Pentene | 8.0 | 2,2,4-Trimethylpentane | 0.9 | n-Propylbenzene | 2.5 |
| cis-2-Pentene | 1.3 | Benzene | 90[a] | Decane | 2.7 |
| Isoprene | 6.7 | Heptane | 1.1 | m- and p-Ethyltoluene | 0.7 |
| 2,3-Dimethylbutane | 2.3 | Methylcyclohexane | 1.1 | 1,3,5-Trimethylbenzene | 1.5 |
| 2-Methylpentane | 7.7 | 2,3,4-Trimethylpentane | 1.3 | o-Ethyltoluene | 1.1 |
| Cyclopentane | 2.4 | 2-Methylheptane | 1.2 | 1,2,4-Trimethylbenzene | 0.6 |
| 3-Methylpentane | 2.1 | 3-Methylheptane | 0.9 | 1,2,3-Trimethylbenzene | 0.9 |
| 1-Hexene | 7.2 | Octane | 3.1 | m-Diethylbenzene | 0.6 |
| Hexane | 1.3 | Toluene | 60[a] | Undecane | 2.6 |
| 2,4-Dimethylpentane | 1.1 | Nonane | 2.8 | p-Diethylbenzene | 0.6 |
| Methylcyclopentane | 0.8 | Ethylbenzene | 3.9 | Dodecane | 2.4 |
| 2-Methylhexane | 1.1 | m- and p-Xylene | 3.5 | | |

**Table 2** Limits of detection (LOD) for a selection of linear, branched and aromatic hydrocarbons on the I/VOC channel of cTAG. Compounds are presented in order of volatility from isopentane (5 carbon atoms) to dodecane (12 carbon atoms). [a] Contamination from the adsorbent materials used in the collector leads to elevated limits of detection for benzene and toluene, consistent with previous reports (Cao and Hewitt, 1994).

may become trapped on the filter, hindering their measurement (Hatch et al., 2017). Such compounds are unlikely to elute on our GC column, which is optimized for less-polar species. During normal operation of the instrument we thus include a filter inline.

**4.5 Region of sensitivity overlap**

Bornyl acetate ($C_{12}H_{20}O_2$) is found in essential oils from pine trees (Garneau et al., 2012). Though it elutes close to tridecane, which is outside the reliably quantifiable range on the SVOC channel, we were able to quantify it on both the I/VOC and SVOC channels in McCall, Idaho during the 2019 FIREX-AQ field campaign. The concentration ranges from 0.1 to 16.5 ppt as measured on the SVOC channel. The correlation of concurrent sampling points on both channels is shown in Fig. 6. The I/VOC and SVOC channels agree on average within 11 %, with $R^2 = 0.79$ and 96 % of the data points agreeing within ±4.8 ppt (3 times the I/VOC channel limit of detection for bornyl acetate). We consider this excellent agreement between the two channels given that the sample sizes differ by more than 2 orders of magnitude, they are calibrated independently, and all data points are within a factor of 10 of the detection limit on the I/VOC channel.

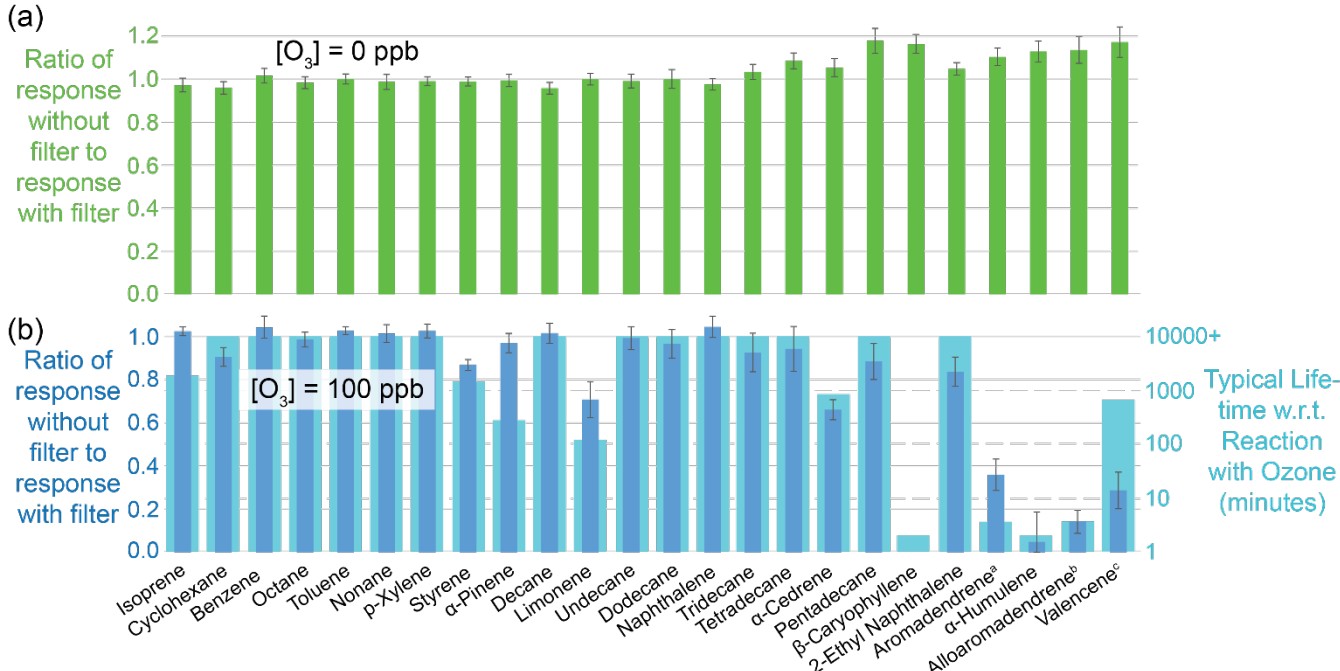

**Figure 5** Results of evaluation of effectiveness of sodium thiosulfate filter in mitigating the effects of ozone on reactive analytes during collection of VOCs and IVOCs. Error bars represent plus or minus one standard deviation of six replicate measurements. Panel (a) confirms that in the absence of ozone, the filter does not affect the detected quantities of each compound, except for slight (< 15 %) losses of IVOCs with the filter present leading to response ratios greater than 1. Panel (b) demonstrates that the filter prevents substantial losses of reactive species in the presence of ozone, confirming that without preventive measures ozone can significantly deplete certain analytes. Values are compared with published lifetimes with respect to reaction with ozone, with overall qualitative agreement. Published lifetimes are from Atkinson and Arey (2003) unless above 10,000 minutes or otherwise indicated. [a] Pollmann et al., 2005; [b] No published experimental value; assumed reactivity equal to that of the stereoisomer aromadendrene; [c] Ham, 2013

## 4.6 Measurements of ambient air

cTAG is sensitive to a wide variety of organic compound classes. Figure 7 shows some sample chromatograms from Berkeley, California ambient air sampled from outside the lab window, the Livermore 2018 deployment (Sect. 3.6) and McCall, Idaho for the 2019 FIREX-AQ field campaign highlighting compounds of interest. The total ion chromatograms and selected ion chromatograms with mass-to-charge ratio of 57, the dominant ion in most alkanes, show the volatility range detected as well as the overlap region of both channels between $C_{14}$ and $C_{16}$ alkane-equivalent volatility. Common air toxics such as BTEX, PAHs and quinones are readily visible, as well as biogenic terpenes and aldehydes, organic acids and polar biomass burning markers. Compounds as polar as glucose (five hydroxy groups) can be detected on the SV channel.

cTAG can observe gas-phase chemicals and many of their oxidation products in gas and particle phases. Naphthalene is a gas phase product of incomplete combustion of fossil fuels (Baek et al., 1991) and in urban areas has been observed to



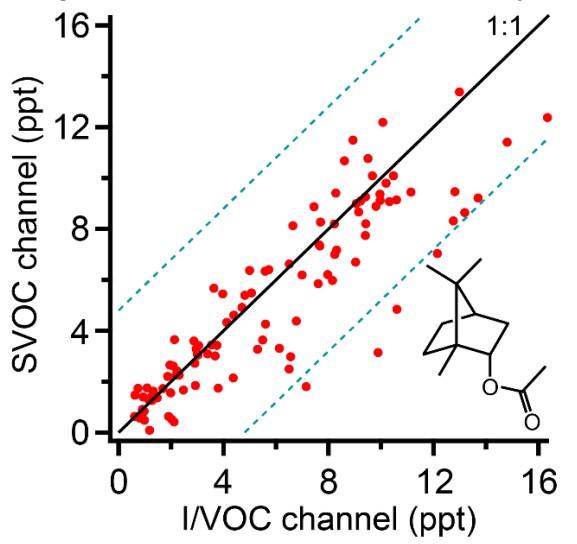

**Figure 6** Bornyl acetate measured on the I/VOC and SVOC channels of cTAG during FIREX-AQ. Dashed lines represent ±4.8 ppt (3 times the I/VOC channel limit of detection for bornyl acetate). 96 % of data points fall within the region bounded by the dashed lines. $R^2$ is 0.79.

primarily originate from vehicle emissions (Howsam and Jones, 1998; Lim et al., 1999). Phthalic anhydride is a major gas-

phase photooxidation product of naphthalene and phthalic acid has been found in secondary organic aerosol formed from naphthalene photooxidation (Chan et al., 2009; Kleindienst et al., 2012; Wang et al., 2007). cTAG is sensitive to naphthalene on the I/VOC channel and the sum of phthalic anhydride and phthalic acid on the SVOC channel (phthalic acid converts to phthalic anhydride on the collection cell). As Fig. 8 shows, the precursor and products have extremely distinct temporal profiles, with naphthalene concentrations elevated at night and in the early morning hours while the phthalic anhydride plus

phthalic acid signal rises in the early to mid-afternoon, consistent with secondary formation and in agreement with observations at other field sites (Williams et al., 2010).

An analysis of straight-chain alkanes along with some aromatics, alkenes and branched alkanes demonstrates cTAG's volatility range and highlights the different source categories for sub-groups of this set of compounds. The cross-correlation matrix (Fig. 9) indicates three distinct groupings of compounds. The first group is characteristic of gasoline emissions,

spanning a volatility range from below $C_5$ alkane-equivalent volatility to about $C_{11}$ (Gentner et al., 2013) and including linear, branched and cyclic alkanes as well as aromatics. It shows short periods of elevated levels in the early morning consistent with a morning rush hour traffic pattern (Fig. 10). The lack of a similar peak near the end of the day could point to the outsized contribution of cold start emissions to total VOC emissions from gasoline vehicles (Drozd et al., 2016, 2019), as the sampling site was located in a primarily residential area. There is also a well-defined group from $C_{20}$ to $C_{26}$ alkanes with a smooth diurnal

variability and daily maximum concentration in the late afternoon (Fig. 10), consistent with a petroleum-based evaporative source such as asphalt (Khare et al., 2020). The afternoon peaks in concentration result in a clear anti-correlation with the rest of the alkanes in Fig. 9 (blue and dark red areas). The third group is less pronounced but follows a roughly similar pattern to


**Figure 7** Example chromatograms from cTAG. Panels (a) and (b) demonstrate the range of volatility covered by the two channels, including their overlap region. Single ion chromatograms (panels (c) through (f)) show examples of the compound classes observable by cTAG, including aromatics, polycyclic aromatic hydrocarbons, organic acids and anhydrosugars.

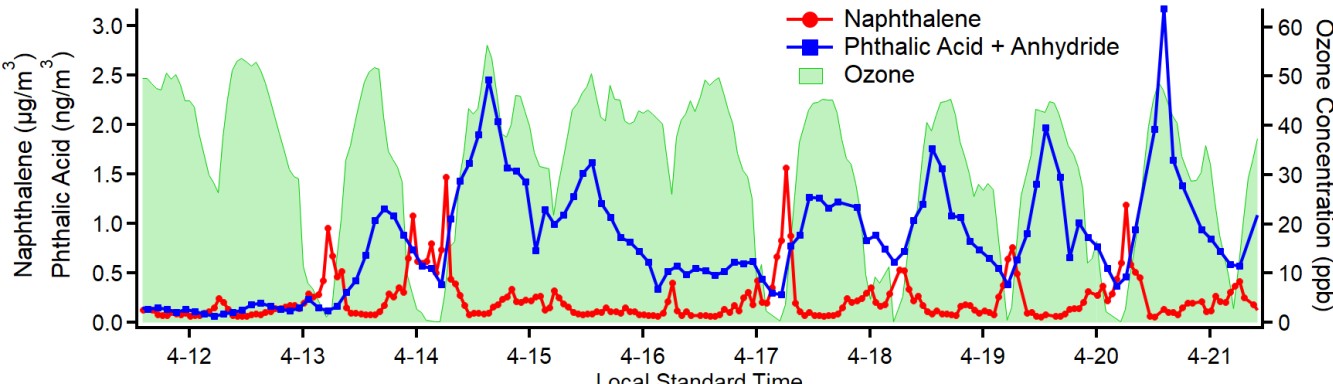

**Figure 8** Concentration timelines for naphthalene, a primary emission from vehicle exhaust measured on cTAG's I/VOC channel, and the sum of phthalic anhydride and phthalic acid, secondary photooxidation products of naphthalene detected on cTAG's SVOC channel. Another secondary compound, ozone, measured at a nearby Bay Area Air Quality Management District monitoring site, provides an independent indication of photochemical activity.

the high-volatility source category and likely corresponds to diesel fuel emissions, which typically span a volatility range from about $C_{10}$ to $C_{22}$, with minor contributions from $C_{23}$–$C_{25}$ alkanes (Gentner et al., 2013; Isaacman et al., 2012; Drozd et al., 2021). Our data suggest that overlapping contributions from gasoline and the petroleum-based evaporative sources make this source group less well-defined in the correlation matrix, but semivolatile compounds that have no other major source still correlate well with each other.

## 5 Summary and concluding remarks

The Comprehensive Thermal Desorption Aerosol Gas Chromatograph is a novel instrument capable of measuring organic compounds from $C_5$ through $C_{32}$ alkane-equivalent volatility on two separate channels connected to a single HRToFMS. For the first time, VOC emissions, reactive intermediates, and secondary products can all be observed on a single instrument at hourly time resolution. The expanded range of measurable compounds allows for more robust source categorization, with detailed chemical specificity of each identified source category.

Building off development of previous TAG family instruments, development for cTAG focused on the I/VOC channel as well as miniature gas chromatographs to enhance field portability of the instrument. The I/VOC collector is optimized for collection of $C_5$ through $C_{16}$ alkane-equivalent volatility organics to capture major biogenic VOC and IVOC emissions and ensure overlap with the SVOC channel collection range. The dynamic dilution system with controlled liquid evaporation developed in house is shown to produce stable, linear calibration curves and allows for maximum flexibility in calibration of the I/VOC channel, including quantification of commercially available VOCs and IVOCs not commonly available in calibration gas cylinders or which are difficult or prohibitively expensive to put in such cylinders. Placement of a $Na_2S_2O_3$ impregnated filter in the sampling path for the I/VOC channel is shown to effectively remove ozone without removing



**Figure 9** Pearson's R correlation matrix for a suite of petroleum-derived compounds observed by cTAG in Livermore, California between April 11th and April 21st, 2018, ordered by volatility. Two distinct groupings emerge from $C_5$ alkane-equivalent volatility to $C_{11}$ and $C_{20}$ to $C_{26}$, with a third less distinct but still prominent grouping from $C_{11}$ to $C_{19}$.

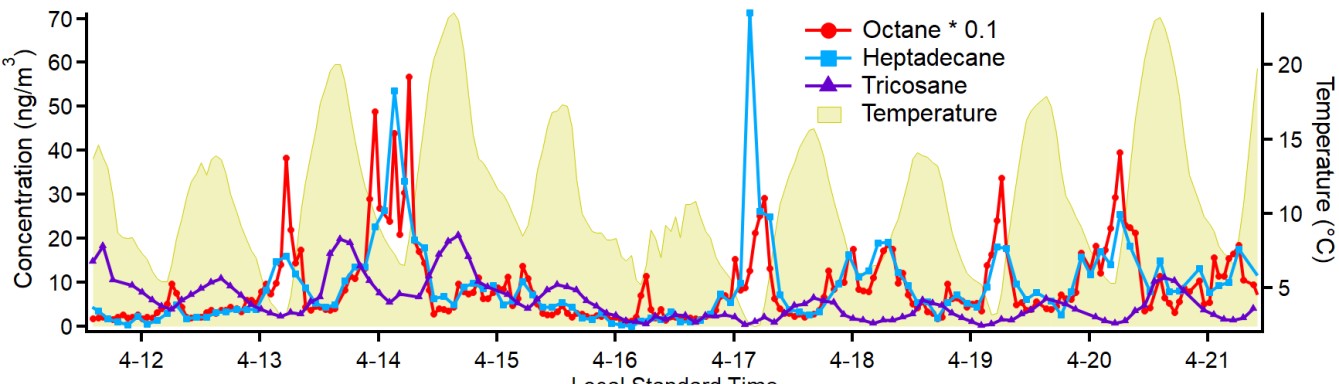

**Figure 10** Timelines for characteristic species from each of the three main groupings of compounds evident in the correlation matrix (Fig. 9). The two more volatile groups, represented by octane and heptadecane, tend to peak in the early morning, consistent with morning rush hour cold start emissions. The least volatile group, which includes tricosane, appears to have an evaporative source as it tracks closely, but slightly delayed, with ambient temperature.

appreciable amounts of analytes of interest. Example chromatograms, timelines and correlations for a suite of compounds at a polluted urban field site demonstrate some of the potential for analysis of the data sets produced by this instrument.

Comprehensive, speciated measurements of reactive organic carbon across the full range of volatility are required to fully understand the atmospheric processes that lead to secondary aerosol formation and dictate the atmospheric lifetimes of key atmospheric oxidants (Heald and Kroll, 2020; Hunter et al., 2017). Moreover, the organic composition and thus the dominant atmospheric processes are likely to vary greatly in different environments (e.g. remote, forested, rural/agricultural, urban), and current science lacks such comprehensive measurements across these different types of sites (Heald and Kroll, 2020). The cTAG is designed for field portability and speciated measurement of a significant fraction of the total reactive organic carbon, making it an ideal choice for helping to close this gap in our current scientific understanding.

## Appendix A Miniature gas chromatogram development

cTAG uses dual miniature gas chromatographs to preserve compactness and independent temperature control. Figure A1 shows a schematic of the final design and a photo of a GC hub on the instrument. Having reproducible temperature ramps is critical for batch chromatogram analysis, since the exact elution time of each compound is dependent upon the column temperature. Figure A2 shows that we can run a consistent, repeatable temperature program using PID control on the mini GCs.


(a)

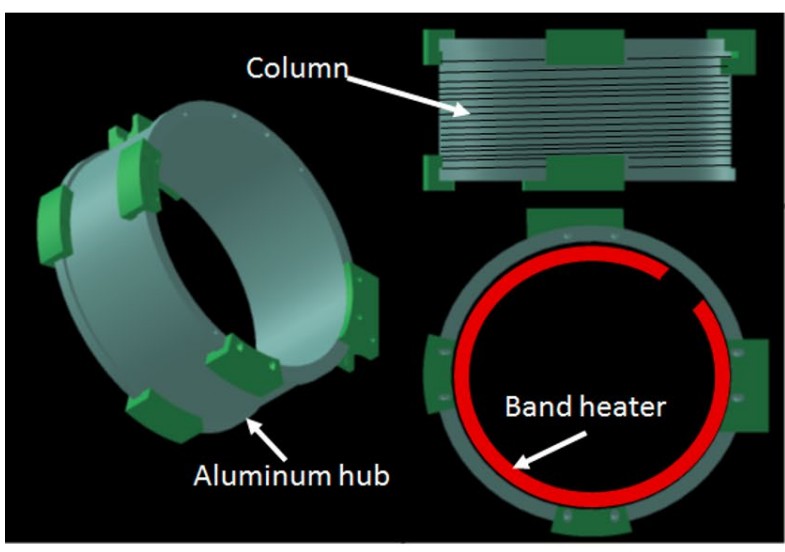

(b)

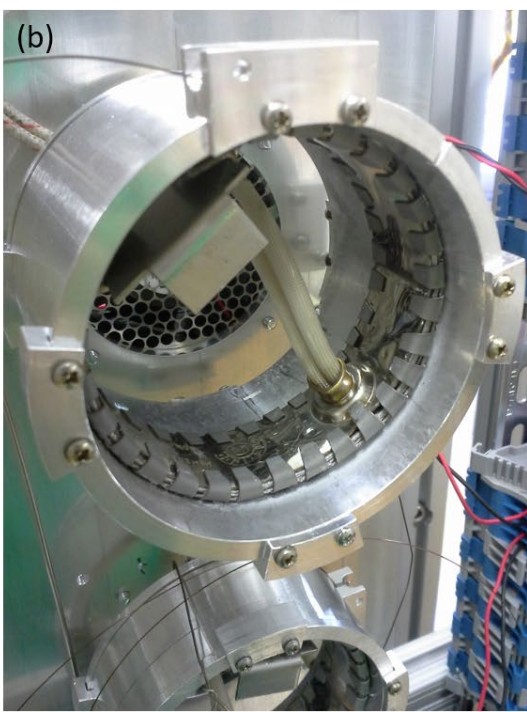

**Figure A1** Schematic and photo of miniature gas chromatographs. Clamps on the edges of the rim of the hub hold the chromatography column in place at each end. An aluminium sheet is wrapped around the outside of the hub over the column to further ensure even heating of the column itself.

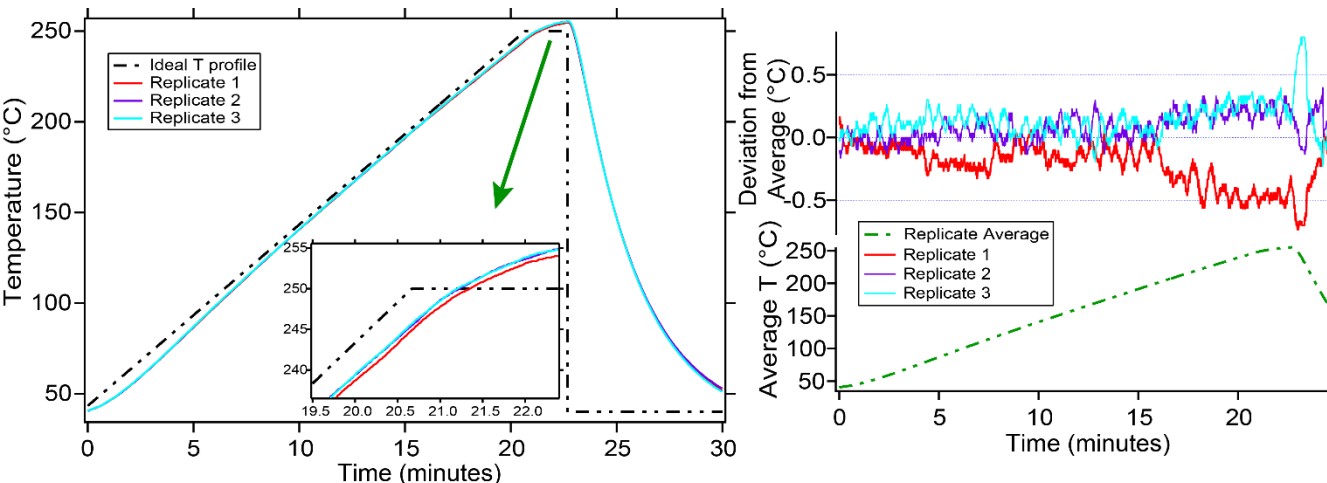

**Figure A2** Demonstration of reproducibility of temperature ramps on the miniature gas chromatographs. This leads to consistent elution times of compounds of interest, streamlining batch peak integration.



*Data availability.* Data for each figure are available from the first author upon request.

*Author contributions.* AG, SH and NK proposed the concept of cTAG. RAW and NK designed the instrument with input from AG and SH. RAW and NK built the instrument. RJW designed and built the dynamic dilution system. JJ provided instrument control hardware and software and customization for their use with cTAG. RAW and YL performed ozone removal evaluation experiments. RAW, NK and AG designed and coordinated the Livermore 2018 deployment and RAW and NK collected data from that deployment. AG, RAW and YL coordinated involvement in the FIREX 2019 deployment and RAW and YL collected data during that deployment. RAW, NK and YL performed data analysis. RAW prepared the manuscript with input and revisions from AG, NK, YL and JJ.

*Competing interests.* The authors declare that they have no conflict of interest.

*Acknowledgements.* Wen Xu and Thorsten Hohaus provided support and customization for the instrument control software. The Livermore Area Recreation and Park District (LARPD) provided the site and infrastructure in Livermore for the Spring 2018 field deployment. The authors would like to especially thank Bruce Aizawa and Steve Sommers of LARPD for enabling us to do measurements at the site. Hourly temperature and ozone measured approximately 100 yards away from our site were provided by the Bay Area Air Quality Management District. Christos Stamatis, Dr. Lindsay Hatch and Dr. Kelley Barsanti from the Barsanti research group at University of California Riverside provided sodium thiosulfate impregnated filters and advised and assisted with ozone testing.

*Financial support.* Instrument development work was supported by the US Department of Energy SBIR/STTR under grant DE-SC0011397. R. Wernis was supported by the US Environmental Protection Agency (EPA) Science To Achieve Results (STAR) Fellowship Assistance Agreement no. FP-91778401-0. FIREX measurements were supported by the National Oceanic and Atmospheric Administration under grant NA16OAR4310107.

*Disclaimer*: This publication has not been formally reviewed by EPA. The views expressed in this publication are solely those of the authors, and EPA does not endorse any products or commercial services mentioned in this publication.



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
