# Peer review of "Development of an *In Situ* Dual-Channel Thermal Desorption Gas Chromatography Instrument for Consistent Quantification of Volatile, Intermediate Volatility and Semivolatile Organic Compounds"

_Atmospheric Measurement Techniques, 2021_

## Author Response (AR1)

**1st reviewer's comments with Inline Responses**

Review of "Development of an In Situ Dual-Channel Thermal Desorption Gas Chromatography Instrument for Consistent Quantification of Volatile, Intermediate Volatility and Semivolatile Organic Compounds" by Wernis et al.:

Summary: The authors describe a novel instrument configuration called the cTAG, building upon previous TAG instruments. The cTAG includes an I/VOC channel with a separate heated GC column, extending the instrument to sample directly emitted VOCs and IVOCs not previously measureable with the same HRToFMS detector. They also describe a new gas calibration system, the success of an ozone removal system, and the performance of the instrument in several field settings. The paper is clear and well written. The addition of the I/VOC channel will be a useful upgrade to the SV-TAG system. I have only a few substantive comments, and some technical comments. I think the authors should clarify the utility of this instrument a bit, as I suggest in my comments below. I suggest publication after these minor revisions.

Main Comments:

Pg 3 Ln 27: The way this sentence is written, it implies the cTAG can measure all precursors and all of their related oxidation products in SOA across 15 decades of volatility, but that is not the case. The GC columns are probably quite limited in which oxidation products they can transmit. And while the GC does provide speciation of isomers, there are other instruments (CIMS or PTR) that can measure precursors and a much wider variety of oxidation products (including in SOA with FIGAERO, or some custom PTR inlets) with which you can infer some of the important speciation by comparing with previous GC studies. While the advances in this paper are certainly quite useful and fill an important niche, I suggest tempering the language here and elsewhere a bit, especially when discussing the abilities to measure the range of oxidation products.

**We appreciate your comment and completely agree with these changes. We have expanded the discussion of the limitations of GC earlier in the introduction section, removed the sentence in question, and modified the remainder of that paragraph as follows: "The Comprehensive Thermal desorption Aerosol Gas chromatograph (cTAG) combines an I/VOC collector based upon the design of Gentner et al. (2012) and one channel of the SV-TAG joined together before a HRToFMS to access a broader volatility range of speciated organic compounds in a single instrument than previously achieved. Nonpolar and some polar VOCs and IVOCs as well as nonpolar and derivatization-amenable polar SVOCs are quantitatively collected, including many primary and secondary organics that lend insight into important sources and oxidation processes in atmospheric chemistry."**

Pg 10 Ln 6: I don't think you need Section 3 as a separate section. When I first read it, I was confused why you weren't showing me any results, e.g. the calibration curves, how the ozone scrubber worked. Then I saw you give the results in basically duplicate sections in Sect. 4. I think it would read much cleaner if you just move the text from Sect. 3 into the corresponding spots in Sect. 4, so it doesn't seem duplicated.

**We agree that the flow of the manuscript is improved by combining these sections and have made this change.**

Pg 18 Ln 14: For a casual reader, it is really hard to identify the three groups of correlated compounds in this plot for two reasons. First, it's just a lot to take in. I think it would be very useful to put boxes around the correlated groups to clearly show the reader where to look, e.g., one box around the top left, one box around the correlations from undecade through phytane, and one box around the bottom right correlations. And second, you're discussing these as volatility groups in the text, but the figure doesn't have any of that volatlility information. Can you include an alkane equivalent volatility scale, like you do with the dashed lines in Fig 7?

**We have made the suggested changes to the figure.**

Pg 20 Ln 12: Here is another example where "organic compounds" is too broad, and I think you will help the reader by specifying the subset of compounds that can be measured, e.g., nonpolar, weakly polar, unoxidized and lightly oxidized, etc.

Pg 20 Ln 14: There are certainly other instruments that can measure VOC emissions, reactive intermediates, and secondary products at greater than hourly resolution on a single instrument. Even if you add the qualifier of "speciated", you should again temper the language by explaining the subset of compounds that can be sampled in relation to what is out there in the atmosphere. I'm not trying to put down the GC measurements, I just think there is no reason to overstate the capabilities. The GC is certainly useful for atmospheric measurements, and the reader will be better served by having a clear idea of the cTAG's capabilities and limitations.

**Thank you for highlighting these two sentences as needing changes. We have modified the language to read "The Comprehensive Thermal Desorption Aerosol Gas Chromatograph is a novel instrument capable of measuring nonpolar and some polar organic compounds as well as some more oxidized semivolatile organics from C5 through C32 alkane-equivalent volatility on two separate channels connected to a single HRToFMS. This set of quantifiable compounds encompasses many key VOC pollutants, reactive intermediates, and secondary products, all captured at hourly time resolution. The expanded range of measurable compounds allows for more robust source categorization, with detailed chemical specificity of each identified source category."**

Technical Comments:

Pg 5 Ln 11: Here you say the glass beads do not trap I/VOCs, but in the Fig. 2 caption you say the IVOCs are captured on the glass beads. Doesn't matter scientifically, but make sure these are consistent.

**Thank you for pointing out this inconsistency. We have edited the Fig. 2 caption removing the mention of glass beads in that sentence.**

Pg 11 Ln 24: When you say "this technique is effective for some sesquiterpenes as well", it reads like you're saying the sodium thiosulfate is effective for removing sesquiterpenes, since the previous sentence was about ozone removal. I believe you mean sodium thiosulfate is effective for removing ozone artifacts while sampling sesquiterpenes? Please reword.

**We have reworded this statement as follows: "Pollmann et al. (2005) demonstrated that this technique is effective at preventing ozone reaction artifacts for some sesquiterpenes as well as the previously studied monoterpenes and hydrocarbons."**

Pg 13 Ln 10: The "10" line number is somehow in the middle of the table.

**Thank you for pointing this out. This looks like a bug in the conversion from MS Word to PDF. We will make sure this issue is fixed for the final submission.**

Pg 13 Ln 24: It would be helpful to quote your usual sample volume here.

**Done.**

Pg 17 Ln 12: Since you're saying here that the data in Fig 7 come from these several locations, you should probably specify here or in the figure caption which panels (a-f) came from which locations.

**We have added the locations in the Fig. 7 caption.**

Pg 17 Ln 16: Need to define BTEX and PAH.

**Done.**

Pg 17 Ln 18: Use "SVOC" instead of "SV" to be consistent.

**Done.**

Pg 18 Ln 7: Do you have a citation or evidence for conversion of phthalic acid to phthalic anhydride on the collection cell? Maybe not necessary but could be good to document.

**We do not have a citation for this, but our group has observed this using an authentic phthalic acid standard and this interpretation has been our standard practice. We have altered the parenthetical statement to read "We have found through laboratory testing of an authentic phthalic acid standard that phthalic acid converts to and is detected as phthalic anhydride in cTAG."**

Pg 19 Ln 1: In Fig. 7, you should switch your labels to read "I/VOC channel" instead of "VOC channel" to be consistent with the earlier text.

**Done, thanks.**

Pg 24 Ln 1: Some journals require all data to be publicly available at the time of publication, e.g., with a doi instead of just upon request. Perhaps the authors will consider adopting this standard even if AMT does not require it, in order to encourage high standards of research in the future.

**Thank you for this suggestion. We will consider adopting this practice in the future, but for this publication our data will remain available upon request, which is sufficient to comply with AMT guidelines.**

**2nd Reviewer's Comments with Inline Responses**

Review of "Development of an In Situ Dual-Channel Thermal Desorption Gas Chromatography Instrument for Consistent Quantification of Volatile, Intermediate Volatility and Semivolatile Organic Compounds" MS No.: amt-2021-156

This paper describes the design and testing of an automated instrument for field measurements of a broad spectrum of organic compounds from VOCs to SVOCs. Given the importance of atmospheric organics, this instrument will be a useful addition to many field studies. The new instrument combines many previous "TAG" technologies developed by this group over 15 years that have been described in previous papers.

The paper describes the instrument, results from experiments to characterize a variety of performance characteristics from compound breakthrough to ozone scrubbing, and some sample field data. The experiments appear to be carefully performed and the paper is clearly written. I recommend the paper be published after the authors have address the following comments.

Main comments

While the instrument is very impressive, it subject to the inherent limitation of sorbents, gas chromatography, and complex sampling systems. While there is derivatization of the SVOC channel, there is none on the I/VOC channel. These limitations inevitably mean that more polar species will be lost, such as multigeneration reaction products. That is fine, every instrument has its limitations, but I agree with reviewer #1 that the review paper needs to provide a more realistic assessment of the strengths and weaknesses of this instrument in the context atmospheric chemistry. While I am sure we will learn a lot from the deployment of this instrument, it inevitably will need to be complimented with other techniques.

The discussion is focused on individual species, which are used as indicators of sources and atmospheric processes (e.g. Figure 10). However, as these authors know well, many lower-volatility species cannot be separate using traditional one-dimensional chromatography. This is clear in the chromatographs shown in Figure 7 which show large "humps" of unresolved complex mixture. The revised paper needs some discussion of this limitation; for example, including an estimate of the fraction of the atmospheric organic mass they believe this instrument is speciating or detecting.

**Thank you for raising this point. We agree that more clarification on the types of compounds this instrument can detect is warranted. We have expanded the discussion of the limitations of GC in the 3rd and 4th paragraphs of the introduction, including predictions of the quantifiable fraction of gas-phase organics and particle-phase organics of the techniques used in this instrument based on prior literature. We have also altered the language used in the final paragraph of the introduction and first paragraph of the summary section to better reflect what this instrument can and cannot measure.**

More polar species likely will be more challenging than hydrocarbons. The example calibration curves (Figure 4) are all for hydrocarbons. It would be helpful to display results for some more polar (challenging) compounds. Is the instrument response as linear and well correlated? There is discussion in the text of instrument performance for one ketone. Figure 5 does not show many sticky compounds.

**Thanks for pointing this out. We hope that changes made in response to your previous comment as well as the other reviewer's first comment partially address your concerns; that is, we are not claiming to be able to measure very polar compounds on the I/VOC channel. On the SVOC channel we are able to see some polar compounds due to the use of online derivatization, but that is not the focus of this paper as it has been documented elsewhere (Isaacman et al., 2014).**

**In Figure 4 the focus is on the gas-liquid comparisons on the I/VOC channel, for which we are limited by what species we had available in a gas cylinder, which were almost entirely hydrocarbons. In Figure 5 the focus is on ozone-reactive compounds.**

Minor comments

In the text you say that glass beads do not collect any organics, but in caption for Figure 2 you say they collect IVOCs. I believe the latter is correct, but the text should be made consistent.

**Thank you for pointing out this inconsistency. Reviewing the breakthrough volume for glass beads (https://www.sisweb.com/index/referenc/glassbed.htm) suggests a negligible amount is retained on them during sampling. E.g. for dodecanol, the least volatile compound in the sisweb table referenced that we would analyze on the I/VOC channel, the breakthrough volume on 10mg of glass beads would be 5 mL, much less than our sample volume of 1 L. We have therefore edited the Fig. 2 caption removing the mention of glass beads in that sentence.**

"Additionally, for benzene all data points had the mean of the 0 ppb values subtracted from them" I found this confusing. Presumably 0 ppb values are when you are running instrument on "zero air" or something similar.

**We have reworded this statement to read "Additionally, for benzene all data points had the mean of the zero air only (0 ppb) values subtracted from them..."**

The evidence of evaporation of tricosane in Figure 10 is a neat result.

**Thanks!**

**List of Changes in Manuscript**
- Merged text from section 4 into corresponding parts of section 3 and adjusted references to section numbers throughout text.
- Pg 2 Ln 31 to Pg 3 Ln 3: Changed from "… but due to the GC temperature ramp and sample collection time has a temporal resolution of 20 minutes to 1 hour." to "However due to the GC temperature ramp and sample collection time GC-based methods have a temporal resolution of 20 minutes to 1 hour. Additionally, typically primary pollutants such as alkanes and aromatics and early generation secondary products such as carbonyls and alcohols are measurable but multifunctional or fully aged species are too thermally unstable or polar to measure by GC-MS. For example, Chung et al. (2003) were able to speciate 55-85% of total VOCs in urban sites in the Los Angeles Basin, with the lower end of the range corresponding to greater photochemical processing and more of the VOC mass present in oxidized species."

- Pg 3 Ln 6: Added "…and detection is limited to compounds for which the ionization reaction with the chosen reagent is energetically favorable."
- Pg 3 Ln 20: Added "…including alkanoic acids, polyols, diacids, sugars and other multifunctional compounds…"
- Pg 3 Ln 24 to Pg Ln 28: Added "Using TAG with an impactor cell, Williams et al. (2010) were able to quantify an estimated 20% of fine ($PM_1$) organic aerosol mass as measured by an aerosol mass spectrometer at an urban site. While this statistic has not been estimated directly for the SV-TAG and likely varies with the measurement location and conditions, the use of online derivatization on that instrument would increase the analyzable fraction of OA mass."
- Pg 4 Ln 3 to end of introduction: Changed from "Until now no single instrument existed that can measure both precursors and their SOA products at the speciated molecular level that spans the relevant 15 decades in volatility. The Comprehensive Thermal desorption Aerosol Gas chromatograph (cTAG) combines a I/VOC collector based upon the design of Gentner et al. (2012) and one channel of the SV-TAG joined together before a HRToFMS to access this entire range of organic volatility at once." to "The Comprehensive Thermal desorption Aerosol Gas chromatograph (cTAG) combines an I/VOC collector based upon the design of Gentner et al. (2012) and one channel of the SV-TAG joined together before a HRToFMS to access a broader volatility range of speciated organic compounds in a single instrument than previously achieved. Nonpolar and some polar VOCs and IVOCs as well as nonpolar and derivatization amenable polar SVOCs are quantitatively collected, including many primary and secondary organics that lend insight into important sources and oxidation processes in atmospheric chemistry."
- Pg 11 Ln 26: Removed "Based on the results of these tests, a final collector composition was chosen with quantities of the more aggressive adsorbents in between those of the second and third prototypes tested."
- Pg 15 Ln 26: Changed from "Pollmann et al. (2005) demonstrated that this technique is effective for some sesquiterpenes as well as the previously studied monoterpenes and hydrocarbons." to "Pollmann et al. (2005) demonstrated that this technique is effective at preventing ozone reaction artifacts for some sesquiterpenes as well as the previously studied monoterpenes and other hydrocarbons."
- Pg 18 Ln 20: Replaced "BTEX, PAHs" with "benzene, toluene, ethylbenzene, xylene, polycyclic aromatic hydrocarbons"
- Pg 20 Ln 2: Replaced "SV" with "SVOC"
- Pg 20 Ln 8: Changed from: "Phthalic acid converts to phthalic anhydride on the collection cell." to "We have found through laboratory testing of an authentic phthalic acid standard that phthalic acid converts to and is detected as phthalic anhydride in cTAG."
- Pg 22 Ln 11: Changed from "The Comprehensive Thermal desorption Aerosol Gas Chromatograph is a novel instrument capable of measuring organic compounds from C5 through C32 alkane-equivalent volatility on two separate channels connected to a single HRToFMS. For the first time, VOC emissions, reactive intermediates, and secondary products can all be observed on a single instrument at hourly time resolution." to "The Comprehensive Thermal Desorption Aerosol Gas Chromatograph is a novel instrument capable of measuring nonpolar and some polar organic compounds as well as some more oxidized semivolatile organics from $C_5$ through $C_{32}$ alkane-equivalent volatility on two separate channels connected to a single HRToFMS. This set of quantifiable compounds encompasses many key VOC pollutants, reactive intermediates, and secondary products, all captured at hourly time resolution."

- Figure 2 caption: Removed "glass beads and"
- Figure 7: Changed "VOC" to "I/VOC" on labels in this figure.
- Figure 7 caption: Added locations where data was collected for each panel in Figure 7.
- Figure 9: Added alkane number markings on top and left axes and indications of the three major groupings of alkanes on the right and bottom of the figure.